# Board Gender Diversity and Carbon Proactivity: The Influence of Cultural Factors

**Haifei Wang** [1], **Qingliang Tang** [2] **and Ting Guo** [3,*]

1   Accounting School, Jilin University of Finance and Economics, Changchun 130117, China
2   School of Business, Western Sydney University, Sydney 1797, Australia
3   Library, Guizhou Institute of Technology, Guiyang 550003, China
*   Correspondence: gting2010@hotmail.com

**Abstract:** Due to inadequate studies, our knowledge of the effect of female directors and national culture on the corporate response to climate change is still limited. To address this gap, the purpose of this paper is to investigate the dynamic relationship between gender diversity on the board of directors and corporate carbon proactivity and how two dimensions of national culture (individualism and indulgence) moderate this relationship. This study focuses on large companies that disclosed carbon-related information via the CDP survey in 2011–2017. Our findings show that gender diversity promotes corporate carbon proactivity. Furthermore, the positive effect of gender diversity on carbon proactivity is weaker when firms are in countries marked by a higher level of individualism and indulgence. As far as we know, this study is the first to explore and document the empirical evidence on the dynamic impact of gender diversity in the corporate governance body and national culture on managers' climate change behaviors in terms of green proactivity.

**Keywords:** board gender diversity; individualism; indulgence; carbon proactivity; climate change

## 1. Introduction

Over the past two decades, the academic literature has paid increasing attention to corporate carbon emissions and carbon accounting. Many studies have considered the consequences of corporate carbon decisions and issues (e.g., carbon disclosure, carbon emissions, and carbon risks) on financial markets. Some studies investigated how the enforcement of climate change regulations impacts market reactions and shareholder returns (Chapple et al. 2013; Jiang and Luo 2018; Luo and Tang 2014; Wang et al. 2021, 2022). Other studies explored the value relevance of carbon emissions or emissions allowances in different countries or regions, such as the United States, the European Union, the United Kingdom, Japan, and Australia (Baboukardos 2017; Choi et al. 2021; Clarkson et al. 2015; Griffin et al. 2017; Matsumura et al. 2014). Furthermore, Nguyen (2018) studied the relationship between carbon risk and firm performance. In contrast, Jung et al. (2018) and Herbohn et al. (2017) examined the effect of a firm's carbon-related risk on lending decisions.

With the development of carbon emissions practices and the proposal of carbon-peak and carbon-neutral targets in all countries around the world, enterprises should respond to and manage climate change from the perspective of corporate governance mechanisms and carbon accounting practices. Several studies investigate whether and how corporate governance affects a firm's carbon disclosures, assurance, and performance. For example, Prado-Lorenzo and Garcia-Sanchez (2010), Liao et al. (2015), Peters and Romi (2014), and Hollindale et al. (2019) examined the effects of characteristics of corporate boards, such as gender diversity, board independence, and the presence of an environmental committee and a chief sustainability officer on promoting information related to greenhouse gases (GHGs).[1] In addition, Datt et al. (2018) found that firms with an environmental committee and carbon reduction incentives tend to adopt external carbon assurance. Furthermore,

Haque (2017) documented that when companies have greater gender diversity on the board, they are inclined to undertake more carbon reduction initiatives. Luo and Tang (2021) reveal that the overall quality of corporate governance mechanisms is positively associated with corporate carbon performance. This positive effect is stronger when firms exhibit greater awareness of carbon risks and adopt a more proactive carbon strategy.

However, few studies have examined how firms' governance mechanisms affect carbon proactivity. Corporate carbon proactivity includes a high level of awareness of climate change, transparent carbon disclosure, advanced carbon management activities, innovative carbon reduction initiatives, and leadership in response to climate change. Our study attempts to fill this gap in the literature by investigating how board gender diversity can play into firms' proactive responses to addressing climate change. Previous literature finds that the presence of female directors on the board is an important corporate governance mechanism that plays a critical role in decision making in the corporate environmental, social, and governance domain (e.g., Bear et al. 2010; Ben-Amar et al. 2017; Liao et al. 2015; McGuinness et al. 2017; Post et al. 2011; Rao and Tilt 2016). Therefore, our study examines the impact of gender diversity on firms' carbon proactivity. In addition, an extensive stream of prior studies documents that national institutions and culture may impact corporate social responsibility (*CSR*) codes, practices, and performance across firms (e.g., Ioannou and Serafeim 2012; Luo and Tang 2016a; Peng and Lin 2009). As a soft and implicit institution, national culture likely influences the perceptions of management concerning the severity and prominence of climate change and thus determines firms' carbon strategy and proactivity. We also investigate whether and how national culture affects the association between gender diversity on the board and carbon proactivity.

Our research sample includes large companies from 16 countries that participated in the CDP (previously known as the Carbon Disclosure Project) survey from 2011 to 2017. Corporate carbon proactivity is measured based on the rank score developed by the CDP, which considers firms' carbon disclosure, awareness, management, and leadership. Eight ordinal values (1–8) are assigned to represent the ranks of D–, D, C–, C, B–, B, A–, and A (the highest rank), respectively. The higher the value, the greater the carbon proactivity. We provide consistent empirical evidence that corporate carbon proactivity is positively affected by board gender diversity. We use Hofstede's framework of national culture and focus on the dimensions of individualism and indulgence (Hofstede et al. 2010). Our results show that the positive effect of board gender diversity on carbon proactivity is weakened by the cultural dimensions of individualism and indulgence.

We make a few contributions to the current literature. First, our study fills a void in the literature on gender diversity on the board and carbon proactivity. Previous studies investigated the link between board gender diversity and *CSR* reporting (e.g., Bear et al. 2010; Ben-Amar et al. 2017; Liao et al. 2015; McGuinness et al. 2017; Post et al. 2011; Rao and Tilt 2016). In the carbon context, most early literature studied the relationship between gender diversity and the disclosure of carbon information (Hollindale et al. 2019; Liao et al. 2015; Peters and Romi 2014; Prado-Lorenzo and Garcia-Sanchez 2010). However, a measure of carbon disclosure only reveals how transparent a firm's carbon information is; it does not reveal what the firm has performed to manage climate change. The carbon proactivity measure in our study is more comprehensive, as it combines green investment with carbon management and assurance. The previous higher disclosure score is different from greater proactivity and represents superior underlying performance a firm has conducted or achieved. Thus, disclosure does not fully reflect the various proactive activities a firm has undertaken to reduce carbon emissions. Our study sheds extra light on the role of gender diversity in promoting firms' overall proactivity in addressing climate change.

Second, previous studies show the direct impact of national culture on firms' environmental, social, and governance decisions (e.g., Luo and Tang 2016a; Peng and Lin 2009; Vitolla et al. 2019). To the best of our knowledge, no studies have examined the moderating effect of national culture on the relationship between board gender diversity and carbon proactivity. Our research tests two dimensions of national culture—individualism and indulgence—and

suggests that the positive relationship between board gender diversity and carbon proactivity is weaker for companies domiciled in countries marked by more individualism and indulgence.

Third, our measure of corporate carbon proactivity is innovative. We use the CDP rank score (2016), which considers four aspects of corporate carbon strategy: Awareness, disclosure, management, and leadership. To the best of our knowledge, the current study is the first to investigate the interactive relationship between board gender diversity, national culture, and corporate carbon proactivity using the CDP carbon rank score approach.

Finally, we consider the influence of natural factors on corporate carbon proactivity. We argue that carbon activity is linked to geographic and climate-related factors that are not at the discretion of corporate directors. For example, in regions with higher temperatures, governments are more likely to adopt a stringent climate policy that will motivate companies to enhance carbon proactivity. Moreover, rising sea levels due to global warming may force businesses to move from lower-lying land to higher ground. Thus, businesses that are situated closer to sea level may be more prone to carbon proactivity. These effects of natural factors have been neglected in prior studies. As we use appropriate proxies to control for these effects, our empirical results are robust and convincing.

The remainder of the paper is structured as follows. Section 2 presents the theoretical background and literature review, and the hypotheses are developed in Section 3. We describe our research design in Section 4, including sample selection, the measurement of variables, and empirical models. In Section 5, we provide the empirical results. Section 6 presents our conclusions and the limitations of the study.

## 2. Literature Review

### 2.1. Board Gender Diversity and CSR

Resource dependency theory holds that an organization is an open system that needs to exchange resources from the outside; at the same time, it indicates that board diversity influences *CSR* (Bear et al. 2010). Thus, gender synergy on the corporate board is believed to be a valuable source of competitive advantage. Board diversity can expand board members' networks and linkages to other firms as a result of the human capital provided by board members (Hillman et al. 2000; Hillman and Dalziel 2003), which may improve the environmental performance of the firm (Ben-Amar et al. 2017; Mallin and Michelon 2011).

Female directors may pay more attention to *CSR* reports and environmental, social, and governance disclosures (Griffin et al. 2021; Hafsi and Turgut 2013; Hillman et al. 2007). Females have the unique characteristics of being cooperative, polite, sympathetic, and empathetic (Griffin et al. 2021; Kramer et al. 2006); thus, female directors tend to care about the natural environment (Stevens 2010). Furthermore, empirical evidence suggests that boards with greater female representation are more stakeholder oriented (Al-Shaer and Zaman 2016; Katmon et al. 2019). When corporations include women directors on their boards, it sends a signal to society that they are oriented toward stakeholders (Ibrahim and Angelidis 1994). They may enhance environmental and social objectives more democratically (Hillman et al. 2002; Hussain et al. 2018) because female directors not only comply better with environmental regulations and policies but also are more prone to participate in *CSR* and charity (Bear et al. 2010; Boyd 1990; Glass et al. 2016; Manita et al. 2018; Post et al. 2011).

Women appear better able to tolerate different opinions, and conflict around carbon-related issues may be mitigated through open discussion and transparent information sharing (Nielsen and Huse 2010). The community-oriented characteristics of female directors not only strengthen a board's understanding of communities' social demands (Hollindale et al. 2019) but also lead to successful corporate environmental practices (Glass et al. 2016; Mun and Jung 2018) because of greater sensitivity to stakeholders' concerns (Horbach and Jacob 2018; Isidro and Sobral 2015). Compared to male directors, female directors appear more participative, cooperative, and democratic (Eagly et al. 2003), and women's leadership style is markedly different from men's (Al-Shaer and Zaman 2016; Bear et al. 2010; Manita et al. 2018). Women leadIn *Proceedings of the* ers pay more attention to stakeholders and are

more effective in corporate social and environmental matters (He and Jiang 2019; Husted 2005).

Critical mass theory as it relates to gender politics states that if women want to make a substantial change to corporate policy, they need to make up 30% of personnel. Galbreath (2010) provides empirical evidence supporting this theory and argues that if the percentage of female directors on the board is too low, they may not determine or change policy. Post et al. (2011) find a positive relationship between female directors and the quality of *CSR* information only when the number of female directors is no less than three. Empirical studies also show that female directors are more likely to support *CSR* activities from ethical initiatives, more effectively helping the company become a good corporate citizen than their male counterparts (Bear et al. 2010; Hafsi and Turgut 2013; Isidro and Sobral 2015; Liao et al. 2015; Rao and Tilt 2016).

Surveys and experiments in the fields of psychology and economics indicate that women tend to score lower on personal achievement and higher on community and public welfare relative to men, who are keener to exercise power and self-direction (Adams and Funk 2012; Schwartz and Rubel 2005). Meanwhile, by enhancing the transparency of information disclosure, female directors can play an essential role on the board (Gul et al. 2011). This is supported by empirical results of a positive relationship between the number of female directors and *CSR* practices (Ben-Amar et al. 2017; Prado-Lorenzo and Garcia-Sanchez 2010). Some studies also show that having more females on the board can decrease the risk of impression management and stimulate sustainable corporate practices and carbon reduction initiatives (Chams and García-Blandón 2019; Nadeem et al. 2017).

### 2.2. National Culture and CSR

Researchers have offered many definitions of national culture, which generally refers to the beliefs, values, codes of conduct, and artifacts shared by the members of a society (Peterson and Barreto 2018). The dimensions of culture are generally quantified in attempts to characterize national culture (e.g., Hofstede 1980; House et al. 2004). Cultural values are typically associated with firms' strategic financial decisions (Chui et al. 2002), economic systems (Kwok and Solomon 2006), and stock price and trading (An et al. 2018; Grinblatt and Keloharju 2001).

Increasingly, studies focus on the relationship between national culture and nonfinancial decisions, such as *CSR* and environmental protection activities (e.g., Gallego-Álvarez and Ortas 2017; Luo et al. 2018; Luo and Tang 2016a; Vitolla et al. 2019; Wagner 2009). In particular, previous studies suggest that *CSR* is an outcome of cultural values and managerial discretion, which may influence the perception of management related to *CSR* matters (e.g., Finkelstein and Hambrick 1996; Wood 1991). Waldman et al. (2006) report that the individualism and power distance dimensions of national culture influence executives' decisions around *CSR*. Husted (2005) indicates that power distance, individualism, and masculinity positively influence a nation's sustainability. Williams (1999) shows that voluntary environmental disclosure correlates with uncertainty avoidance and masculinity.

Similarly, feminine societies tend to focus on social goals, such as protecting the physical environment (Van der Laan Smith et al. 2005). A "feminine" society refers to the overlapping of gender roles between men and women in society and believes that both men and women should be modest, gentle, and concerned about the quality of life. People will see "income, praise, ambition, and challenge" written on the masculinity pole; on the femininity side, it says "peace, cooperation, routine, and security". Buhr and Freedman (2001) find that Canada tends to have higher environmental disclosure than America; the former is collectivist and the latter is individualist from the viewpoint of national culture. Individualism arises from a society in which individuals are rather loosely connected, and each person in the society tends to focus only on himself or his immediate family; as its opposite, Collectivism comes from a strong cohesive society, in which people are integrated from birth into a close-knit group that continuously protects in exchange for the absolute loyalty of each member. Furthermore, literature on the incentives behind corporate carbon reduction is emerging.

Luo and Tang (2016a) find that national culture has a significant direct effect on corporate disclosure of carbon information. Yet there is still limited knowledge of how national culture interacts with corporate governance to jointly affect firms' carbon decisions.

## 3. Hypothesis Development

### 3.1. Board Gender Diversity and Carbon Proactivity

We argue that greater gender diversity can enhance firms' carbon proactivity. According to resource dependency theory, the unique experiences and skills of female directors help companies access various external resources and strengthen corporate carbon proactivity (Ben-Amar and McIlkenny 2015; Mallin et al. 2013) through the sharing of essential information and open communication. For example, Hollindale et al. (2019) argue that women directors can increase a board's communication of community issues through value attunement. In this way, female representation on the board ensures that alternative considerations of climate change are offered (Bear et al. 2010).

Moreover, female directors, who are more democratic and participative, tend to care more about the natural environment than male directors (Eagly et al. 2003; Luo and Tang 2016a; Orij 2010). This encourages more open conversations and information sharing among board members (Eagly et al. 2003; Eagly and Johnson 1990) and increases board effectiveness around climate-related actions (Nielsen and Huse 2010). Owing to their community orientation, female directors are more sensitive to societal stakeholders' interests, including climate change (Glass et al. 2016; Mallin and Michelon 2011). The prior literature provides empirical evidence that female directors are more proactive concerning sustainability initiatives, such as corporate low-carbon strategies (Bear et al. 2010; Glass et al. 2016; Liao et al. 2015; Mallin and Michelon 2011; Rao and Tilt 2016).

Thus, having greater female representation on the board promotes a company's awareness of climate change and stimulates the use of low-carbon technologies to mitigate GHG emissions. Furthermore, a higher percentage of female directors reflects effective consideration of stakeholders' demands (Ben-Amar et al. 2017; Liao et al. 2015). Moreover, Srinidhi et al. (2011) argue that female board members can improve board monitoring and oversight, ensuring that carbon strategies are implemented properly with genuine activities and adequate resources instead of just superficial slogans (Nielsen and Huse 2010). Therefore, we propose the following hypothesis:

**H1.** *Ceteris paribus, there is a positive association between board gender diversity and carbon proactivity.*

### 3.2. The Moderating Effect of National Individualism

According to Matten and Moon (2008), the hypothesis about society and business may derive from different cultural systems. Crossland and Hambrick (2011) show that several cultural characteristics directly influence the decisions of management, which they conceptualize as executive action latitude (Hambrick and Finkelstein 1987). Thus, we would expect gender diversity to influence the ways in which firms respond to climate change differently across distinct societies with different cultures. Because climate change is highly uncertain, there is significant variation in corporate carbon proactivity across countries (Luo et al. 2012; Luo and Tang 2016a). Corporate carbon proactivity is influenced by national culture, and different dimensions may have different effects.

National culture has obvious effects on various organizational practices (Hofstede 1980), especially when formal institutions are weak or absent (Nakpodia et al. 2018). Prior empirical results show that moderating effects exist between national culture and firm innovation (Van Everdingen and Waarts 2003; Wu et al. 2019), corporate entrepreneurship (Turró et al. 2014), and corporate finance and governance outcomes (Hooghiemstra et al. 2015; Lewellyn and Muller-Kahle 2020). As an important part of the *CSR* strategy, carbon proactivity is a complex business decision related to environmental accountability (Hambrick and Finkelstein 1987; Smith et al. 2002). We focus here on the moderating effects of two dimensions of national culture (individualism and indulgence) on the relationship between board gender diversity and corporate carbon proactivity. We consider the moderating effect of individualism first.

In individualist cultures, individuals focus on their own interests and objectives over those of the group (Hofstede 1980). The philosophy behind individualism focuses on self-interest. The higher the individualism, the looser the society. In contrast, in collectivist groups, people care more about public welfare.

In a business environment, the accomplishment of corporate objectives can affect stakeholder judgment (Freeman 1984). Thus, corporate managers deal with *CSR* by participating in stakeholder activities. In individualist societies, the linkages between individuals are weaker and need driven (Hofstede 2001). People prioritize self-actualization over social concerns. In collectivist societies, individuals who desire lifelong relationships are loyal to in-groups. Organizations in collectivist cultures are more cooperative, participatory, and consultative (Buhr and Freedman 2001). The prior literature shows a negative link between individualism and *CSR* performance (Halkos and Polemis 2017; Ho et al. 2012), because collectivist cultures are more sensitive to stakeholder interests. Companies are more prone to disclose social responsibility information in collectivist societies than in individualist societies. Hence, the positive effect of board gender diversity on carbon proactivity may be reduced for firms in societies characterized by more individualism. We thus expect that the influence of board gender diversity on carbon proactivity will be weaker when firms operate in more individualist societies. Therefore, we propose the following hypothesis:

**H2.** *Higher levels of the cultural dimension of individualism weaken the positive relationship between board gender diversity and carbon proactivity.*

### 3.3. The Moderating Effect of National Indulgence

The cultural dimension of indulgence (vs. restraint) refers to the degree to which a society is free to satisfy its desires instead of controlling its impulses through strict societal norms (Hofstede et al. 2010). Indulgence reflects the ability to control personal desires and wishes. Indulgent societies emphasize the importance of freedom of speech and personal life and allow individuals to enjoy life and have fun (Hofstede et al. 2010). In indulgent cultures, individuals tend to spend money and enjoy leisure, and they pay more attention to everyday enjoyment. In contrast, in restrained cultures, individuals try to control their impulses and desires. Until now, few studies have empirically shown the effects of indulgence on corporate carbon proactivity. Indulgent societies are more wasteful and extravagant, which increases environmental pollution. Whereas Halkos and Polemis (2017) find a positive relationship between indulgence and the disclosure of *CSR* information, Gallego-Álvarez and Ortas (2017) show a negative relationship between indulgence and *CSR* practices.

Climate change is marked by high uncertainty but has many negative outcomes for the environment, such as extreme weather, glacial melting, and rising sea levels (Stern 2006). Carbon proactivity is a long-term consideration in that people must control their current wasteful behavior although it may satisfy their current desires and wishes. Simultaneously, the willingness of society to seek out long-term results will influence the adoption of proactive carbon decisions and the innovation of green technologies. Individuals' behavior is inevitably influenced by national culture. Corporate directors in more indulgent societies are prone to be more indulgent. They tend to spend money to satisfy stakeholders' present desires and ignore social welfare, such as investment in *CSR*. A proactive carbon strategy to mitigate the effects of climate change seems more compatible with a national culture characterized by less indulgence. This is because directors, particularly female directors, in less indulgent societies may opt to save money for more proactive carbon reduction initiatives, providing additional assurances and promising more stable future weather. In this way, the positive effect of board gender diversity on carbon proactivity may be mitigated for firms in more indulgent societies. We thus expect that the positive effect of board gender diversity on carbon proactivity will be weaker when firms operate in more indulgent societies. According to the analysis, we propose our final hypothesis as follows:

**H3.** *Higher levels of the cultural dimension of indulgence weaken the positive relationship between board gender diversity and carbon proactivity.*

## 4. Research Design

### 4.1. Sample Selection

Our sample comes from the CDP database. As a leading nongovernmental organization headquartered in the United Kingdom, the CDP collects comprehensive carbon information from the largest companies worldwide. The use of CDP data has two main advantages. First, the CDP uses standardized questionnaires and guidelines for all participating firms. This uniformity ensures consistency in companies' carbon reports over time and means that data are more comparable spatially (Luo et al. 2012). Second, the CDP collects different types of carbon information relevant to different stakeholders' decision making, including information on carbon initiatives and assurance, carbon emissions and reduction, carbon risk and opportunity, and carbon strategy and performance. It is one of the most comprehensive and credible sources of carbon data in the world. The CDP database has been widely used not only for academic study by scholars worldwide but also for commercial purposes by institutional investors (e.g., Bui et al. 2020; Griffin et al. 2017; Ioannou et al. 2016; Kolk et al. 2008; Matsumura et al. 2014; Qian and Schaltegger 2017; Reid and Toffel 2009).

Our initial sample includes the largest companies that respond to the CDP survey from 2011 to 2017. We first deleted companies with missing carbon rank scores, missing scores for dimensions of national culture, and missing financial data. Next, we excluded countries with fewer than 30 observations. Ultimately, we obtained a total of 2731 observations across 16 countries or regions covering 11 Global Industry Classification Standard sectors with firm level and year time. The sample selection process is presented in Table 1.

**Table 1.** Sample Selection Process.

| Initial Sample | Observations |
| --- | --- |
| Firms in the CDP database over the period 2011–2017 that both responded disclosure score and rank score | 5482 |
| Less: | |
|    Observations with incomplete data | 2578 |
|    Countries with fewer than 30 observations | 173 |
| Final sample | 2731 |

### 4.2. Measurement of Carbon Proactivity

We measured corporate carbon proactivity using the carbon rank score assigned to the firm by the CDP based on its responses to the annual CDP questionnaire. This rank score ranges from D to A (Guo et al. 2020), where a higher rank indicates greater carbon proactivity. Specifically, a grade of D means the company only discloses carbon information. A grade of C means the company not only discloses its GHG emissions and activities but is also aware of climate-related opportunities and risks. A grade of B indicates that the company has undertaken carbon reduction activities to manage its carbon emissions and risks and maximize its carbon opportunities. Finally, a grade of A shows that the company demonstrates leadership in carbon awareness, higher-quality disclosure of carbon information, and superior carbon governance and management (CDP Scoring Methodology 2018).

The CDP scoring methodology consists of three steps. In the first step, the CDP calculates points for carbon disclosure and awareness based on the company's responses to the questionnaire. Next, the total points the company has been awarded out of the maximum available points is multiplied by 100. Finally, this score is rounded to the nearest whole number. Points for the management and leadership levels are weighted. Because the four levels are sequential, companies must achieve a score of at least 80% in the previous level to move to the next level and earn a higher ranking (CDP Scoring Methodology 2018, p. 6).

Here we converted the CDP's carbon rank score (D– to A) to an ordinal scale (1–8), where 1 = D– and 8 = A. These values are used to measure carbon proactivity (the primary dependent variable in this study). A higher value represents greater carbon proactivity. In 2016, the CDP adopted a new carbon ranking system. Therefore, we transformed disclosure scores to rank scores for 2011 to 2015. The method used for this transformation is shown

in Table 2. For example, if a company's disclosure score for 2012 is 82, it is transformed to rank B, and the corresponding rank score is 6.

**Table 2.** Process for converting the four consecutive levels from fractions to ranks.

| Level | Percentage Score | 2016 Green Rank | 2011–2015 Disclosure Score | Carbon Proactivity Rank |
|---|---|---|---|---|
| Disclosure | 0–44% | D– | $\leqq 39$ | 1 |
| | 45–79% | D | 40–49 | 2 |
| Awareness | 0–44% | C– | 50–59 | 3 |
| | 45–79% | C | 60–69 | 4 |
| Management | 0–44% | B– | 70–79 | 5 |
| | 45–79% | B | 80–89 | 6 |
| Leadership | 0–79% | A– | 90–99 | 7 |
| | 80–100% | A | 100 | 8 |

Source: CDP Scoring Introduction (CDP Scoring Methodology 2018, p. 6).

*4.3. Empirical Model*

We used ordered multivariate logistic (OML) regression to estimate coefficients as specified in Equation (1) to test H1. We then introduced two interaction variables to test H2 and H3, respectively.

$$CARPRO = \beta_0 + \beta_1 FEMALE + \beta_2 SIZE + \beta_3 LEV + \beta_4 ROA + \beta_5 CSR + \beta_6 LAW + \beta_7 ETS$$
$$+\beta_8 LNGDP + \beta_9 TEMP + \beta_{10} DISTANCE + \sum YEARDUMMY + \sum SECTORDUMMY + \varepsilon \tag{1}$$

Firm-level data with year and industry fixed effect.

$$CARPRO = \beta_0 + \beta_1 FEMALE + \beta_2 IDV + \beta_3 FEMALE * IDV + \beta_4 SIZE + \beta_5 LEV + \beta_6 ROA$$
$$+\beta_7 CSR + \beta_8 LAW + \beta_9 ETS + \beta_{10} LNGDP + \beta_{11} TEMP + \beta_{12} DISTANCE$$
$$+\sum YEARDUMMY + \sum SECTORDUMMY + \varepsilon \tag{2}$$

Firm-level data with year and industry fixed effect.

$$CARPRO = \beta_0 + \beta_1 FEMALE + \beta_2 INDULGENCE + \beta_3 FEMALE * INDULGENCE + \beta_4 SIZE$$
$$+\beta_5 LEV + \beta_6 ROA + \beta_7 CSR + \beta_8 LAW + \beta_9 ETS + \beta_{10} LNGDP + \beta_{11} TEMP + \beta_{12} DISTANCE$$
$$+\sum YEARDUMMY + \sum SECTORDUMMY + \varepsilon \tag{3}$$

Firm-level data with year and industry fixed effect.

*CARPRO* is the dependent variable (carbon proactivity). *FEMALE*, which is the percentage of female board members, represents the firm's board gender diversity. The individualism index (*IDV*) and indulgence index (*INDULGENCE*) identified and developed by Hofstede et al. (2010) are used as the primary test variables. *FEMALE×IDV* represents the interaction between individualism and board gender diversity, and *FEMALE×INDULGENCE* represents the interaction between indulgence and board gender diversity.

We considered three types of control variables. First, we controlled for firm-level factors that may influence carbon proactivity. Because large companies have the capability to do better *CSR* and are under greater social pressure (e.g., Ioannou et al. 2016), we controlled for firm size (*SIZE*), which is measured as the natural logarithm of total sales. Firm leverage (*LEV*) may restrict funding for carbon abatement projects, and thus it was controlled as well. The effect of financial resources on ethical activities and carbon decisions was represented by return on assets (*ROA*; Luo et al. 2013). Furthermore, a corporate board's *CSR* orientation (*CSR*) may influence carbon proactivity practices; *CSR* was a dummy variable that equals 1 if the company has a *CSR* committee and 0 otherwise.

Second, we considered the impact of different institutional factors on corporate carbon behavior at the national level. The previous literature shows that the type of legal system in a country is related to the stakeholder or shareholder orientation of the society and thus affects *CSR*. Therefore, we controlled for the legal system (*LAW*) at the national level (Simnett et al.

2009). *LAW* equals 1 if the firm is headquartered in a region with a civil law system and 0 otherwise. In addition, as a major instrument for regulating climate change policy, the emissions trading scheme (*ETS*) is crucial to the impact of corporate carbon policy. Finally, the level of economic development in a region is closely related to its environmental policies, so we used the gross domestic product per capita (*LNGDP*) to control for the influence of the country's economic development on carbon proactivity (Luo et al. 2013).

Third, we considered the impact of natural conditions on corporate carbon policies. Geographic factors are outside of managers' control, but nevertheless, they have an impact on corporate carbon strategy that should not be ignored. Regions with higher temperatures, for example, may be more committed to reducing carbon emissions. As global warming melts glaciers and sea levels rise, regions closer to the sea will pay more attention to climate issues and resulting corporate carbon strategies. Thus, we included two geographic control variables: The mean annual temperature of the country (*TEMP*) and the distance from the center of the country to the nearest coastline (*DISTANCE*). Finally, the effects of sector and year were considered (e.g., Choi and Luo 2021; Choi et al. 2021; Luo et al. 2012). Figure 1 provides the definitions of variables.

| Variable | Definition | Source |
|---|---|---|
| **Dependent variable** | | |
| *CARPRO* | Rank A = 8, Rank A– = 7, Rank B = 6, Rank B– = 5, Rank C = 4, Rank C– = 3, Rank D = 2, Rank D– = 1 | CDP |
| **Independent variable** | | |
| *FEMALE* | The percentage of female directors on the board | DataStream |
| **Interaction variables** | | |
| *IDV* | Individualism index | Hofstede et al. |
| *INDULGENCE* | Indulgence index | (2010) |
| **Control variables** | | |
| **Country-level variables** | | |
| *LAW* | An indicator variable that equals 1 if the firm is headquartered in a common law country and 0 otherwise | La Porta et al. (1998) |
| *ETS* | An indicator variable that equals 1 if the firm operates in a country that has an operating national emissions trading scheme and 0 otherwise | Website |
| *LNGDP* | The natural logarithm of the gross domestic product per capita | World Bank |
| **Firm-level variables** | | |
| *SIZE* | The natural logarithm of total sales | DataStream |
| *LEV* | Total debt (measured as short-term debt, the current portion of long-term debt, and long-term debt) divided by total assets of book value | DataStream |
| *ROA* | Net income before extraordinary items/preferred dividends, divided by total assets | DataStream |
| *CSR* | A dummy variable that equals 1 if the company has a board-level corporate social responsibility committee and 0 otherwise | DataStream |
| Sector | A dummy variable | DataStream |
| Year | A dummy variable | DataStream |
| **Geographic variables** | | |

**Figure 1.** *Cont.*

| | | |
|---|---|---|
| *TEMP* | The mean annual temperature in the country (in degrees Celsius) | World Bank |
| *DISTANCE* | The distance from the center of the country to the nearest coast (in kilometers) | https://www.pdx.edu/econ/country-geography-data (accessed on 12 August 2022) |

**Figure 1.** Definitions and Sources of Variables. Note: Hofstede et al. (2010); La Porta et al. (1998).

## 5. Empirical Results

### 5.1. Descriptive Analysis

Descriptive statistics for the variables are shown in Table 3. The mean *CARPRO* is 5.955, which implies that our sample firms' average corporate carbon proactivity is 5.96 out of 8. The mean *FEMALE* is 20.008, which suggests the average percentage of female board members in our study sample. Concerning the independent variables, the mean *IDV* and *INDULGENCE* are 77.074 and 60.488, respectively. Regarding country-level variables, approximately 56% of the sample companies are from common-law countries that are shareholder oriented, 34.7% of countries participate in a national emissions trading scheme, and the mean *LNGDP* is 10.791. Furthermore, the mean *SIZE* is 16.18, which suggests our sample firms are among the world's largest companies, comparable to prior empirical results (e.g., Guo et al. 2020; Lemma et al. 2019; Luo and Tang 2021). *ROA* is 6.131% and *LEV* is 26.028% in our sample. Regarding geographic factors that may affect corporate carbon proactivity, the mean *TEMP* is 8.713 and *DISTANCE* is 512.696. The descriptive statistics for the primary variables indicate that our sample is representative.

Table 4 shows the distribution of the sample by sector (Panel A), year (Panel B), and country (Panel C). Panel A shows that of the 11 sectors represented by sample firms, industrials (17.87%), financials (15.82%), and consumer discretionary (12.78%) account for the most firms. Energy (3.37%), telecommunications (3.41%), and real estate (4.03) account for the least. Panel B reveals the sample size by year. The number of firms responding to the CDP increases from 355 (in 2011) to 473 (in 2016), then drops to 297 in 2017. This shows that overall, corporate awareness of carbon proactivity is increasing. Meanwhile, the year-by-year data show that the mean carbon proactivity trends upward over the study period, increasing from 4.721 in 2011 to 6.141 in 2017. Panel C shows the distribution of sample firms by country. Of the 16 countries represented, the United States (34.05%), the United Kingdom (14.87%), and Japan (13.15%) have the highest proportion of firms. Compared to the other countries, these three countries are more industrialized and pay more attention to low-carbon environmental protection.

Table 5 shows the correlations between the major independent variables and the dependent variable of the rank score. Regarding the independent variables of cultural dimension, individualism and indulgence are significantly correlated. The correlations between the other variables indicate that multicollinearity is not a problem in the sample.

**Table 3.** Descriptive Statistics (N = 2731).

| Variable | Mean | SD | P25 | Median | P75 | Min | Max |
|---|---|---|---|---|---|---|---|
| *CARPRO* | 5.955 | 1.507 | 5 | 6 | 7 | 1 | 8 |
| *FEMALE* | 20.008 | 11.900 | 11.11 | 20 | 27.27 | 0 | 50 |
| *IDV* | 77.074 | 16.001 | 68 | 89 | 91 | 46 | 91 |
| *INDULGENCE* | 60.488 | 12.339 | 48 | 68 | 68 | 30 | 78 |
| *SIZE* | 16.185 | 1.348 | 15.277 | 16.271 | 17.108 | 12.642 | 18.906 |
| *CSR* | 0.923 | 0.267 | 1 | 1 | 1 | 0 | 1 |
| *LEV* | 26.028 | 15.891 | 14.39 | 24.71 | 36.23 | 0 | 72.85 |
| *ROA* | 6.083 | 5.993 | 2.03 | 4.98 | 9.09 | −11.56 | 26.54 |
| *ETS* | 0.347 | 0.476 | 0 | 0 | 1 | 0 | 1 |
| *LAW* | 0.560 | 0.496 | 0 | 1 | 1 | 0.000 | 1.000 |
| *LNGDP* | 10.791 | 0.220 | 10.64 | 10.817 | 10.923 | 10.246 | 11.390 |
| *TEMP* | 8.713 | 4.182 | 7.734 | 9.031 | 11.085 | −6.220 | 22.360 |
| *DISTANCE* | 512.696 | 560.167 | 33.237 | 116.085 | 1148.69 | 13.743 | 1712.840 |

Notes: Please see Figure 1 for detailed definitions and sources of variables.

**Table 4.** Distribution of the Sample by Sector, Year, and Country.

**Panel A: Distribution by GICS Sector (N = 2731)**

| Sector | Energy | Materials | Industrials | Consumer Discretionary | Consumer Staples | Health Care | Financials | Information Technology | Telecommu Nications | Utilities | Real Estate | Total |
|---|---|---|---|---|---|---|---|---|---|---|---|---|
| N | 92 | 281 | 488 | 349 | 278 | 181 | 432 | 281 | 93 | 146 | 110 | 2731 |
| Percent | 3.37 | 10.29 | 17.87 | 12.78 | 10.18 | 6.63 | 15.82 | 10.29 | 3.41 | 5.35 | 4.03 | 100 |

Notes: GICS, Global Industry Classification Standard.

**Panel B: Distribution by Year (N = 2731)**

| YEAR | 2011 | 2012 | 2013 | 2014 | 2015 | 2016 | 2017 | Total |
|---|---|---|---|---|---|---|---|---|
| N | 355 | 380 | 382 | 405 | 439 | 473 | 297 | 2731 |
| Percent | 13 | 13.91 | 13.99 | 14.83 | 16.07 | 17.32 | 10.88 | 100 |
| *CARPRO* | 4.721 | 5.292 | 5.848 | 6.291 | 6.943 | 6.180 | 6.141 | 5.955 |

Notes: Please see Figure 1 for detailed definitions and sources of variables.

**Panel C: Means by Country**

| Country | N | Percent |
|---|---|---|
| Australia | 54 | 1.98 |
| Canada | 104 | 3.81 |
| Denmark | 41 | 1.5 |
| Finland | 88 | 3.22 |
| France | 142 | 5.2 |
| Germany | 140 | 5.13 |
| Ireland | 36 | 1.32 |
| Italy | 64 | 2.34 |
| Japan | 359 | 13.15 |
| Netherlands | 49 | 1.79 |
| Norway | 33 | 1.21 |
| Spain | 72 | 2.64 |
| Sweden | 85 | 3.11 |
| Switzerland | 128 | 4.69 |
| United Kingdom | 406 | 14.87 |
| United States | 930 | 34.05 |
| Total | 2731 | 100 |

Notes: The third column represents the percentage of firms out of the full sample in a particular country.

**Table 5.** Correlations (N = 2731).

| Variable | CARPRO | FEMALE | IDV | INDULGENCE | SIZE | CSR | LEV | ROA | ETS | LAW | LNGDP | TEMP | DISTANCE |
|---|---|---|---|---|---|---|---|---|---|---|---|---|---|
| CARPRO | 1.000 | | | | | | | | | | | | |
| FEMALE | 0.176 | 1.000 | | | | | | | | | | | |
| IDV | −0.053 | 0.318 | 1.000 | | | | | | | | | | |
| INDULGENCE | −0.060 | 0.241 | 0.772 | 1.000 | | | | | | | | | |
| SIZE | 0.212 | 0.065 | −0.025 | −0.098 | 1.000 | | | | | | | | |
| CSR | 0.082 | 0.041 | −0.037 | −0.070 | 0.147 | 1.000 | | | | | | | |
| LEV | 0.058 | 0.021 | 0.019 | −0.077 | −0.037 | 0.061 | 1.000 | | | | | | |
| ROA | −0.025 | 0.101 | 0.189 | 0.221 | −0.054 | 0.005 | −0.058 | 1.000 | | | | | |
| ETS | 0.191 | −0.029 | −0.178 | −0.226 | 0.279 | 0.096 | 0.165 | −0.048 | 1.000 | | | | |
| LAW | −0.075 | 0.072 | 0.855 | 0.715 | −0.049 | −0.041 | 0.001 | 0.160 | −0.230 | 1.000 | | | |
| LNGDP | −0.052 | 0.122 | 0.352 | 0.552 | 0.052 | −0.043 | −0.074 | 0.134 | −0.131 | 0.211 | 1.000 | | |
| TEMP | 0.050 | −0.186 | −0.180 | −0.385 | −0.001 | 0.004 | 0.128 | −0.002 | 0.157 | −0.183 | −0.298 | 1.000 | |
| DISTANCE | −0.052 | 0.057 | 0.598 | 0.475 | 0.120 | 0.001 | 0.001 | 0.068 | −0.191 | 0.683 | 0.328 | −0.394 | 1.000 |

Notes: Pearson (Spearman) correlation coefficients are below (above) the diagonal. Please see Figure 1 for detailed definitions and sources of variables.

### 5.2. Multivariate Regression Analysis

Table 6 reports our main regression results based on Equations (1)–(3). For all specifications, we use OML regression and control for year and industry fixed effects. Column (1) contains only our independent variable board gender diversity (*FEMALE*) and the control variables. Columns (2) and (3) add the interaction variables *FEMALE×IDV* and *FEMALE×INDULGENCE* to the main model to test the moderating effects of the cultural dimensions of individualism and indulgence on the association between gender diversity and carbon proactivity.

**Table 6.** Results of OML Regression Based on Board Gender Diversity.

| VARIABLE | (1) | (2) | (3) |
|---|---|---|---|
| | *CARPRO* | *CARPRO* | *CARPRO* |
| *FEMALE* | 0.0171 *** | 0.0584 *** | 0.0376 *** |
| | (5.12) | (3.12) | (2.85) |
| *IDV* | | −0.00449 | |
| | | (−0.83) | |
| *FEMALE×IDV* | | −0.0518 ** | |
| | | (−2.14) | |
| *INDULGENCE* | | | 0.0155 ** |
| | | | (1.97) |
| *FEMALE×INDULGENCE* | | | −0.0378 * |
| | | | (−1.67) |
| *SIZE* | 0.352 *** | 0.357 *** | 0.359 *** |
| | (10.93) | (11.04) | (11.07) |
| *CSR* | 0.721 *** | 0.719 *** | 0.734 *** |
| | (5.07) | (5.06) | (5.14) |
| *LEV* | 0.000388 | 0.000198 | 0.000449 |
| | (0.16) | (0.08) | (0.18) |
| *ROA* | 0.0119 * | 0.0120 * | 0.0112 |
| | (1.76) | (1.77) | (1.64) |
| *ETS* | 0.709 *** | 0.720 *** | 0.707 *** |
| | (7.96) | (8.06) | (7.93) |
| *LAW* | −0.0980 | 0.235 | −0.268 * |
| | (−0.96) | (1.24) | (−1.75) |
| *LNGDP* | 0.0221 | 0.142 | −0.176 |
| | (0.12) | (0.74) | (−0.78) |
| *TEMPERATURE* | −0.00103 | 0.00627 | 0.00354 |
| | (−0.10) | (0.60) | (0.33) |
| *DISTANCE* | −0.000200 ** | −0.000191 * | −0.000161 |
| | (−2.00) | (−1.91) | (−1.56) |
| SECTOR EFFECT | YES | YES | YES |
| YEAR EFFECT | YES | YES | YES |
| N | 2731 | 2731 | 2731 |

Notes: Ordered multivariate logistic (OML) regression is used to test the impact of board diversity (the percentage of female directors on the board) on carbon proactivity and the impact of the interaction of individualism and indulgence on carbon proactivity. T statistics are in parentheses. *, **, and *** represent significance at $p < 0.10$, $p < 0.05$, and $p < 0.01$.

In Column (1), we see that the coefficient of *FEMALE* is positive and significant at the 1% level, which confirms H1 that companies with a higher percentage of female directors have better carbon proactivity. This result is compatible with Liao et al. (2015), who documented that companies are prone to voluntarily disclose carbon information when there is more gender diversity on the board. Regarding the control variables, *SIZE* and *ROA* are positively linked to carbon proactivity, which indicates that larger firms and more profitable firms are more likely to engage in carbon proactivity. However, the relationship between *LEV* and carbon proactivity is not significant in our sample. The coefficient of *CSR* is positive and significant at the 1% level, which suggests that boards with a stakeholder orientation have greater carbon proactivity. Consistent with our expectations, carbon regulations such as

emissions trading schemes (*ETS*) have a positive effect on carbon proactivity. The coefficients of *LAW*, *LNGDP*, and *TEMP* are negative but not significant. One possible reason for this is that cultural factors may have a more substantial effect on carbon proactivity than economic factors. The coefficient of *DISTANCE* is negative and significant at the 5% level, which indicates that the shorter distance from the center of the country to the nearest coast, the greater the climate change proactivity, which suggests that geographic factors influence carbon proactivity.

The empirical results in Column (2) show that the interaction effect of *FEMALE×IDV* is negative and significant at the 5% level, which strongly supports H2 that the positive effect of board gender diversity on carbon proactivity decreases as individualism increases. Previous empirical literature shows that in collectivist societies, citizens value social responsibilities (e.g., in waste recycling) over self-actualization, and collectivist cultures are more cooperative, participatory, and consultative (Buhr and Freedman 2001). In regions with less individualism, corporate managers may feel more responsible for the broader social welfare and thus take on more corporate social responsibilities related to stakeholders.

In Column (3), we see that the interaction effect of *FEMALE×INDULGENCE* is negative and significant at the 10% level, which provides empirical evidence for H3. This indicates that female directors' positive impact on carbon proactivity is weaker for firms in countries high in indulgence. Regions with higher scores for indulgence adopt relatively little carbon proactivity in response to climate change, which may be because low-carbon behaviors consume many social and economic resources (restructuring the local industrial chain, etc.), which is not in line with the ideals of a local culture that values current consumption and enjoys life in the present.

*5.3. Robustness Tests*

We ran a series of robustness tests to check our main findings. First, our sample only contains firms that voluntarily disclose carbon information to the CDP (Matsumura et al. 2014). To address this self-selection bias, we use Heckman's (1979) two-stage estimation. In the first stage, we use maximum likelihood to design a Probit model to estimate a company's likelihood of disclosing carbon emissions information. The Probit model is also a generalized linear model. When the dependent variable is a categorical variable, there are four commonly used analysis models: The Linear probability model (LPM), Logistic Model, Probit Model, and Log-linear model. In the same way as Logistic regression, Probit regression is also divided into binary Probit regression, ordered multiclassification Probit regression, and unordered multi-classification Probit regression.

$$\begin{aligned} DIS \;=\; & \beta_0 + \beta_1 FEMALE + \beta_2 SIZE + \beta_3 LEV + \beta_4 ROA + \beta_5 CSR + \beta_6 LAW \\ & + \beta_7 ETS + \beta_8 LNGDP + \beta_9 TEMP + \beta_{10} DISTANCE + \beta_{11} POPIN \\ & + \sum YEARDUMMY + \sum SECTORDUMMY + \varepsilon \end{aligned} \tag{4}$$

Firm-level data with year and industry fixed effect.

In Equation (4), *DIS* equals 1 if the firm participates in the CDP and 0 otherwise. In the second stage, we perform OML regression controlling for year and industry effects. The empirical results (not shown) support our findings.

Second, U.S. firms account for as much as 34.05% of our sample. To address bias due to the uneven distribution among countries, we remove U.S. companies from the research sample and repeat the regression. Again, the results are not significantly different from our baseline findings. Third, instead of using OML regression in Model (1), we use Tobit regression to test our hypotheses, as the dependent variable ranges from 1 to 8. In the process of regression, sometimes continuous explained variables can only select a certain range of values because they are truncated or censored, which will lead to inconsistent estimators. Davidson and MacKinnon (2004) defined that if some observed values are systematically removed from the sample, it is called truncation. When no observations are eliminated, but some are limited to a certain point, this is called blocking. The Tobit regression results show that our main inferences remain unchanged. Fourth, because firms'

growth may influence carbon proactivity, we replace *ROA* with an alternative measure of profitability, *TOBINQ*, which equals the total market value at the end of the year. *TOBINQ* represents firms' growth opportunities and is a longer-term index than *ROA*. Our results are still valid. Finally, we add *Scope 1* and *Scope 2*, which are measured as the logarithm of total absolute Scope 1 and Scope 2 emissions, as control variables in our model. Scope 1 and Scope 2 emissions are significant GHG emissions that may affect corporate carbon proactivity. The results suggest that our main inferences are still qualitatively the same.

*5.4. Further Analyses*

Finally, we perform further analyses of subsamples of firms based on membership in a carbon-intensive sector, government policies, and legal system. First, we test whether our main results vary according to firms' membership in a carbon-intensive sector (*INTENSIVE*). *INTENSIVE* equals 1 if the firm operates in a carbon-intensive sector (such as energy, materials, or utilities) and 0 otherwise (e.g., Choi and Luo 2021; Choi et al. 2021; Luo et al. 2012). Table 7 shows the results for gender diversity, the two cultural dimensions of individualism and indulgence, and carbon proactivity based on membership in a carbon-intensive sector. Our main inferences only hold for firms in carbon-intensive sectors, which suggests that board gender diversity plays a more prominent role in promoting carbon proactivity in carbon-intensive sectors than in non-carbon-intensive sectors. The moderating effects of individualism and indulgence also only exist for firms in carbon-intensive sectors, which implies that the relative effects of these cultural dimensions on carbon proactivity depend on firms' membership in a carbon-intensive sector.

Second, we test whether the direct effect of board gender diversity varies between subsamples in countries with and without an emissions trading scheme. An emissions trading scheme is a government mechanism that establishes a market so firms can trade emissions allowances or rights. It sets a price for carbon that internalizes the cost of emissions for participating firms. *ETS* equals 1 if the country in which the firm is located has a national emissions trading scheme and 0 otherwise. An emissions trading scheme may amplify or diminish the effect of culture on carbon proactivity. The coefficients of the main variables in Table 8 are all significant and the same as our main results for both subsamples, except for the *FEMALE×IDV* coefficient in the subsample in countries without an emissions trading scheme.

Finally, there is a consensus among researchers that firms in countries with a common-law system tend to be under less pressure from shareholders to engage in carbon proactivity than firms in countries with a code-law system (and a stakeholder orientation; Hope 2003; Jaggi and Low 2000; Jiang and Luo 2018; Luo and Tang 2016b; Zhou et al. 2016). The variable *LAW* equals 1 if the company is in a country with a common-law system and 0 otherwise. The empirical results in Table 9 show that the coefficient of *FEMALE* is positive and significant at the 1% level for firms in countries with a common-law system; we do not find any significant results for firms in countries with a code-law system. Concerning the interaction effect, *FEMALE×IDV* is positively related to carbon proactivity at the 5% level. This means that the effect of gender diversity on corporate carbon proactivity is sensitive to the country's legal system.

**Table 7.** Results of subsample analyses based on membership in a carbon-intensive sector.

| Variable | (1) INTENSIVE = 1 CARPRO | (2) INTENSIVE = 1 CARPRO | (3) INTENSIVE = 1 CARPRO | (4) INTENSIVE = 0 CARPRO | (5) INTENSIVE = 0 CARPRO | (6) INTENSIVE = 0 CARPRO |
|---|---|---|---|---|---|---|
| FEMALE | 0.0370 *** | 0.143 *** | 0.0906 *** | 0.0140 *** | 0.0329 | 0.0221 |
| | (4.51) | (3.22) | (2.85) | (3.80) | (1.52) | (1.47) |
| IDV | | 0.0129 | | | −0.00435 | |
| | | (0.94) | | | (−0.72) | |
| FEMALE×IDV | | −0.150 ** | | | −0.0223 | |
| | | (−2.44) | | | (−0.81) | |
| INDULGENCE | | | 0.0288 * | | | 0.0128 |
| | | | (1.69) | | | (1.43) |
| FEMALE×INDULGENCE | | | −0.103 * | | | −0.0159 |
| | | | (−1.77) | | | (−0.63) |
| SIZE | 0.489 *** | 0.514 *** | 0.496 *** | 0.343 *** | 0.344 *** | 0.347 *** |
| | (6.17) | (6.41) | (6.23) | (9.72) | (9.74) | (9.80) |
| CSR | 0.743 | 0.791 * | 0.958 ** | 0.664 *** | 0.665 *** | 0.664 *** |
| | (1.62) | (1.67) | (2.00) | (4.41) | (4.42) | (4.41) |
| LEV | 0.0000126 | −0.00158 | −0.00128 | −0.00130 | −0.00131 | −0.00109 |
| | (0.00) | (−0.21) | (−0.17) | (−0.52) | (−0.52) | (−0.43) |
| ROA | 0.00306 | 0.00111 | 0.000181 | 0.0104 * | 0.0105 * | 0.00955 |
| | (0.22) | (0.08) | (0.01) | (1.74) | (1.75) | (1.58) |
| ETS | 0.644 *** | 0.567 *** | 0.577 *** | 0.750 *** | 0.753 *** | 0.755 *** |
| | (3.26) | (2.70) | (2.88) | (7.36) | (7.37) | (7.40) |
| LAW | −0.479 ** | −0.198 | −0.695 ** | 0.0340 | 0.238 | −0.154 |
| | (−2.14) | (−0.49) | (−2.12) | (0.29) | (1.09) | (−0.88) |
| LNGDP | −0.961 ** | −0.826 * | −1.093 ** | 0.207 | 0.281 | −0.00568 |
| | (−2.30) | (−1.85) | (−2.16) | (1.05) | (1.33) | (−0.02) |
| TEMP | −0.0115 | 0.00719 | −0.00294 | 0.00789 | 0.0116 | 0.0134 |
| | (−0.58) | (0.34) | (−0.14) | (0.67) | (0.95) | (1.07) |
| DISTANCE | 0.000424 * | 0.000440 * | 0.000449 * | −0.000336 *** | −0.000327 *** | −0.000288 ** |
| | (1.70) | (1.76) | (1.76) | (−2.99) | (−2.91) | (−2.45) |
| SECTOR EFFECT | YES | YES | YES | YES | YES | YES |
| YEAR EFFECT | YES | YES | YES | YES | YES | YES |
| N | 519 | 519 | 519 | 2212 | 2212 | 2212 |

Notes: Ordered multivariate logistic regression is used to test the impact of board diversity (the percentage of female directors on the board) on carbon proactivity and the impact of the interaction of individualism and indulgence on carbon proactivity. T statistics are in parentheses. *, **, and *** represent significance at $p < 0.10$, $p < 0.05$, and $p < 0.01$.

**Table 8.** Results of subsample analyses based on participation in a national emissions trading scheme.

| Variable | (1) ETS = 1 | (2) ETS = 1 | (3) ETS = 1 | (4) ETS = 0 | (5) ETS = 0 | (6) ETS = 0 |
|---|---|---|---|---|---|---|
| | CARPRO | CARPRO | CARPRO | CARPRO | CARPRO | CARPRO |
| FEMALE | 0.0122 ** | 0.0784 ** | −0.0259 | 0.0220 *** | 0.0454 * | 0.0744 *** |
| | (2.19) | (2.34) | (−1.15) | (5.09) | (1.92) | (4.03) |
| IDV | | −0.0198 ** | | | 0.00390 | |
| | | (−2.14) | | | (0.55) | |
| FEMALE×IDV | | −0.0775 * | | | −0.0312 | |
| | | (−1.71) | | | (−1.04) | |
| INDULGENCE | | | −0.00772 | | | 0.0301 *** |
| | | | (−0.58) | | | (2.89) |
| FEMALE×INDULGENCE | | | −0.0736 * | | | −0.0894 *** |
| | | | (−1.73) | | | (−2.99) |
| SIZE | 0.449 *** | 0.461 *** | 0.450 *** | 0.293 *** | 0.296 *** | 0.304 *** |
| | (7.12) | (7.23) | (7.05) | (7.53) | (7.59) | (7.78) |
| CSR | 0.910 ** | 0.814 ** | 0.854 ** | 0.675 *** | 0.685 *** | 0.688 *** |
| | (2.55) | (2.29) | (2.38) | (4.26) | (4.31) | (4.33) |
| LEV | 0.00338 | 0.00282 | 0.00414 | −0.00200 | −0.00218 | −0.00197 |
| | (0.70) | (0.59) | (0.85) | (−0.67) | (−0.73) | (−0.66) |
| ROA | 0.0225 * | 0.0260 * | 0.0217 | 0.00865 | 0.00838 | 0.00806 |
| | (1.71) | (1.95) | (1.62) | (1.07) | (1.04) | (1.00) |
| ETS | −0.384 ** | 0.486 | −0.521 * | 0.0775 | 0.0977 | −0.220 |
| | (−2.07) | (1.46) | (−1.92) | (0.61) | (0.41) | (−1.13) |
| LAW | −0.0831 | 0.307 | −0.254 | 0.172 | 0.162 | −0.157 |
| | (−0.25) | (0.86) | (−0.61) | (0.78) | (0.69) | (−0.56) |
| LNGDP | −0.0161 | −0.0000154 | −0.0127 | 0.00977 | 0.0128 | 0.0172 |
| | (−0.74) | (−0.00) | (−0.57) | (0.83) | (1.03) | (1.35) |
| TEMP | −0.000234 | −0.000264 | −0.000157 | −0.000193 | −0.000182 | −0.000126 |
| | (−1.19) | (−1.33) | (−0.77) | (−1.60) | (−1.50) | (−0.99) |
| DISTANCE | 0.0122 ** | 0.0784 ** | −0.0259 | 0.0220 *** | 0.0454 * | 0.0744 *** |
| | (2.19) | (2.34) | (−1.15) | (5.09) | (1.92) | (4.03) |
| SECTOR EFFECT | YES | YES | YES | YES | YES | YES |
| YEAR EFFECT | YES | YES | YES | YES | YES | YES |
| N | 947 | 947 | 947 | 1784 | 1784 | 1784 |

Notes: Ordered multivariate logistic regression is used to test the impact of board diversity (the percentage of female directors on the board) on carbon proactivity and the impact of the interaction of individualism and indulgence on carbon proactivity. T statistics are in parentheses. *, **, and *** represent significance at $p < 0.10$, $p < 0.05$, and $p < 0.01$.

**Table 9.** Results of subsample analyses based on legal system.

| Variable | (1) LAW = 1 CARPRO | (2) LAW = 1 CARPRO | (3) LAW = 1 CARPRO | (4) LAW = 0 CARPRO | (5) LAW = 0 CARPRO | (6) LAW = 0 CARPRO |
|---|---|---|---|---|---|---|
| FEMALE | 0.0212 *** | −0.291 ** | −0.358 | 0.00597 | 0.0161 | 0.0252 |
| | (3.60) | (−2.26) | (−0.88) | (1.29) | (0.44) | (1.51) |
| IDV | | −0.0871 ** | | | −0.00747 | |
| | | (−2.37) | | | (−0.85) | |
| FEMALE×IDV | | 0.350 ** | | | −0.00844 | |
| | | (2.42) | | | (−0.16) | |
| INDULGENCE | | | −0.241 | | | 0.00312 |
| | | | (−1.59) | | | (0.29) |
| FEMALE×INDULGENCE | | | 0.554 | | | −0.0383 |
| | | | (0.93) | | | (−1.18) |
| SIZE | 0.300 *** | 0.307 *** | 0.291 *** | 0.573 *** | 0.574 *** | 0.575 *** |
| | (7.08) | (7.21) | (6.79) | (10.45) | (10.47) | (10.46) |
| CSR | 0.875 *** | 0.884 *** | 0.879 *** | 0.196 | 0.188 | 0.189 |
| | (4.62) | (4.66) | (4.64) | (0.88) | (0.84) | (0.84) |
| LEV | 0.00427 | 0.00408 | 0.00399 | −0.000130 | −0.000235 | 0.000155 |
| | (1.27) | (1.21) | (1.18) | (−0.03) | (−0.06) | (0.04) |
| ROA | 0.00510 | 0.00619 | 0.00440 | 0.00899 | 0.00961 | 0.0123 |
| | (0.62) | (0.75) | (0.54) | (0.65) | (0.69) | (0.88) |
| ETS | 0.558 *** | 0.560 *** | 0.566 *** | 0.806 *** | 0.810 *** | 0.800 *** |
| | (4.30) | (4.31) | (4.34) | (6.27) | (6.29) | (6.21) |
| LAW | 0.163 | −0.125 | −0.413 | 0.000202 | 0.133 | −0.0215 |
| | (0.22) | (−0.15) | (−0.47) | (0.00) | (0.42) | (−0.06) |
| LNGDP | 0.0158 | 0.0241 | 0.0330 | −0.0665 *** | −0.0607 ** | −0.0836 *** |
| | (0.90) | (1.00) | (1.45) | (−2.61) | (−2.33) | (−2.98) |
| TEMP | −0.000204 | −0.000121 | −0.000118 | 0.00495 *** | 0.00490 *** | 0.00480 *** |
| | (−1.09) | (−0.53) | (−0.57) | (5.89) | (4.84) | (5.40) |
| DISTANCE | 0.0212 *** | −0.291 ** | −0.358 | 0.00597 | 0.0161 | 0.0252 |
| | (3.60) | (−2.26) | (−0.88) | (1.29) | (0.44) | (1.51) |
| SECTOR EFFECT | YES | YES | YES | YES | YES | YES |
| YEAR EFFECT | YES | YES | YES | YES | YES | YES |
| N | 1530 | 1530 | 1530 | 1201 | 1201 | 1201 |

Notes: Ordered multivariate logistic regression is used to test the impact of board diversity (the percentage of female directors on the board) on carbon proactivity and the impact of the interaction of individualism and indulgence on carbon proactivity. T statistics are in parentheses. **, and *** represent significance at $p < 0.05$, and $p < 0.01$.

**6. Conclusions and Limitations**

Our study shows that female board members play an essential role in firms' carbon proactivity and exogenous informal institutions, such as national culture, when it comes to their influence on carbon productivity. After we control for factors such as the legal system, carbon regulations, and the natural environment, our empirical findings reveal the extent to which management's proactive response to climate change depends on gender diversity on the corporate board. More specifically, we find that the cultural dimensions of individualism and indulgence have a significant conditioning influence on board gender diversity and corporate carbon proactivity. This study enhances the understanding of female directors' role in corporate response to climate change under different cultural environments.

Our findings have a range of implications for policymakers, managers, and other stakeholders. Policymakers should consider cultural factors when advocating climate policies and regulations. Our results also suggest that corporations should pay more attention to the unique cultural backgrounds of local managers to ensure a more environmentally friendly board of directors. Such a team can help companies adopt a net-zero business model. Furthermore, the satisfaction of various stakeholders may bring many advantages to companies, such as reputation, low cost of debt, market share, and others.

It is true that our study has some limitations. First, our research sample includes the largest companies in each country that responded to the CDP survey. Future researchers need to study whether these conclusions hold for small and medium-sized companies. Second, we only tested the cultural dimensions of individualism and indulgence. Follow-up research can use our method to examine Hofstede's other four dimensions (power distance, masculinity, long-term orientation, and uncertainty avoidance). In addition, future research can use alternative measures of national culture, for example, Global Leadership and Organizational Behavior Effectiveness, to investigate the effect of national culture on corporate carbon strategy.

**Author Contributions:** Methodology, H.W.; Formal analysis, H.W.; Investigation, T.G.; Resources, Q.T.; Data curation, T.G.; Writing—original draft, H.W.; Writing—review & editing, Q.T.; Visualization, T.G.; Supervision, Q.T.; Funding acquisition, H.W. All authors have read and agreed to the published version of the manuscript.

**Funding:** This research was funded by Humanities and Social Sciences Project of Ministry of Education [grant numbers 22YJA790063].

**Institutional Review Board Statement:** Not applicable.

**Informed Consent Statement:** Not applicable.

**Data Availability Statement:** Not applicable.

**Conflicts of Interest:** The authors declare no conflict of interest.

**Note**

[1] The terms *emissions*, *carbon emissions* and equivalent, and *GHG emissions* are used interchangeably in this paper.

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
