# Peer review of "Board Gender Diversity and Carbon Proactivity: The Influence of Cultural Factors"

_jrfm, doi:10.3390/jrfm16020131_

Round 1

Reviewer 1 Report

I don't think control variable TEMP is appropriate. This variable should measure temperature deviation from a multi-year trend rather than temperature level. Apart from this remark, I congratulate the authors on the interesting research and the clear structure of the paper

Reviewer 2 Report

The paper assesses the dynamic relationship between gender diversity on the board of directors and corporate carbon proactivity and how two dimensions of national culture moderate this relationship. I have read the complete paper and I hold following observation on it.

1.      The research design needs to be improved, information on the data needs to be added as using the firms level and time. In addition definition of probit and tobit and the logic behind using them as robustness check could been further explained.

2.      Equations (1,2, 3 & 4), need to be revised. For instance, more clarity need to be added regarding the data by introducing subscriptions for the variables. E.g., adding  (i) subscript for firm level data or if other subscript needs to be added.

3.      For more clarity for readers, the detailed definitions and sources of variables table in appendix A should have been provided before the other tables or in even in the main text.

4.      More clarity on measurement of the control/independent variables are needed. For instance, LEV variable is measured as Total debt divided by total assets. However, more clarification is needed on whether the short-term debt or the current portion of long-term debt, or long-term debt were used as a proxy for total debt. And here you could add a recent and related work that have used similar firm level variables such as LEV, Firm size (see, Regression Analysis of Macroeconomic Conditions and Capital Structures of Publicly Listed British Firms. Mathematics 202210, 1119.) in the manuscript.

5.      More clarity on the results of probit and tobit and interpretations could be added.

6.      I feel that the paper could gain a lot in terms of readability if some discussion on company-specific details could be added to the discussion and conclusion sections. In fact, companies are the objects of the study of the authors while numbers representing them are rather an effect produced by them.

7.      The notes below the Table 9, need to be revised, there is no variable at 0.1 level (* ).

I wish you best of luck.

Reviewer 3 Report

It is an interesting paper analyzing the relationship between diversity in the board of directors and carbon proactivity. This is a creative topic with not much related research. It is a valid research hypothesis as it is conceivable that the background of the board of directors might impact their decisions.

I think that the paper adds to the existing literature by combining diversity and carbon policy proactivity. There is a very large body of research in these two topics separately but not much, as far as I am aware, combining these two topics.

The paper is not formatted in the usual MDPI template but in the standard (blank) template but this does not affect content. I like that the authors have out effort in some of the classification such as using the well-known (respected) Laporta classification for legal system.

The paper is well structured and the methodology is clear and easy to follow but I still have some comments:

1)       The models seem reasonable i.e., equation 1, but I think that the authors should explain it better. Questions such as: why were those variables selected?

2)       The authors correctly mentioned the potential for self-reporting bias skewing data (self-reporting firms are likely to be doing a better job than non-reporting ones). I think that it would be good to expand this section a bit explaining more about the robustness analysis and actions to reduce the impact of biases.

3)       The list of 16 countries are highly industrialized countries, basically European countries + US and Japan. Arguably there is going to be a high degree of cultural homogeneity (particularly among European countries) that might lead to results that are not extrapolable to other countries (emerging markets). I think that the author should address this concerns (at least from a qualitative point of view).

4)       Some of the expressions need to be more carefully defined. For example “feminine societies tend to focus on social goals” (citing existing literature) requires a more precise definition of what is a feminine society. Similar concerns when dealing with expressions such as “collectivist and the latter is individualist from the viewpoint of national culture”. I am not arguing that these expressions are wrong just that they require more precise definitions as it could have very different meanings for different readers. Any way to roughly quantify this?

5)       Another consideration to take into account, particularly when analyzing gender diversity are family ties with the management/shareholder. A typical example is a female board of director, which increase diversity, but it is the daughter/granddaughter of the management/shareholder, arguably reducing diversity benefits.  
